# Low-Resolution ADCs for Two-Hop Massive MIMO Relay System under Rician Channels

**DOI:** 10.3390/e23081074

**Published:** 2021-08-19

**Authors:** Shujuan Yu, Xinyi Liu, Jian Cao, Yun Zhang

**Affiliations:** 1College of Electronic and Optical Engineering & College of Microelectronics, Nanjing University of Posts and Telecommunications, Nanjing 210023, China; yusj@njupt.edu.cn (S.Y.); 18260039728@163.com (J.C.); y021001@njupt.edu.cn (Y.Z.); 2Institute of Data Science, The Fu Foundation School of Engineering and Applied Science, Columbia University, New York, NY 10027, USA

**Keywords:** MIMO relay system, Rician channel, low-resolution ADCs, achievable sum rate

## Abstract

This paper works on building an effective massive multi-input multi-output (MIMO) relay system by increasing the achievable sum rate and energy efficiency. First, we design a two-hop massive MIMO relay system instead of a one-hop system to shorten the distance and create a Line-of-Sight (LOS) path between relays. Second, we apply Rician channels between relays in this system. Third, we apply low-resolution Analog-to-Digital Converters (ADCs) at both relays to quantize signals, and apply Amplify-and-Forward (AF) and Maximum Ratio Combining (MRC) to the processed signal at relay $R_{1}$ and relay $R_{2}$ correspondingly. Fourth, we use higher-order statistics to derive the closed-form expression of the achievable sum rate. Fifth, we derive the power scaling law and achieve the asymptotic expressions under different power scales. Last, we validate the correctness of theoretical analysis with numerical simulation results and show the superiority of the two-hop relay system over the one-hop relay system. From both closed-form expressions and simulation results, we discover that the two-hop system has a higher achievable sum rate than the one-hop system. Besides, the energy efficiency in the two-hop system is higher than the one-hop system. Moreover, in the two-hop system, when quantization bits q = 4, the achievable sum rate converges. Therefore, deploying low-resolution ADCs can improve the energy efficiency and achieve a fairly considerable achievable sum rate.

## 1. Introduction

In the field of wireless communication, multi-input multi-output (MIMO) systems have been widely used for their superior performance like increasing channel capacity and improving user anti-interference performance [1]. However, MIMO systems also have the following disadvantages. First, when the distances between users and targets are large, the signal cannot reach the targets directly due to the heavy shadow and path loss [2,3,4,5]. Therefore, there is no Line-of-Sight (LOS) between users and targets and only Rayleigh channels can be applied between [6,7]. Second, when a large number of transmitting and receiving antennas are equipped with high-resolution Analog-to-Digital Converters (ADCs), the system will consume tremendous amounts of energy. To be specific, if a high-resolution ADC is with b-bit precision and the sampling frequency is $f_{s}$, $f_{s}*2^{b}$ conversions will be required per second, which means the energy consumption of the system increases exponentially with the quantization accuracy [2].

With the development of multi-hop communication, we can decrease the distances between relays so that LOS can appear. Then we can apply Rician channels instead of Rayleigh channels to reduce the achievable sum rate. Besides, using low-resolution ADCs instead of high-resolution ADCs can alleviate the energy consumption burden of the MIMO system and achieve a fairly considerable achievable sum rate at the same time.

### 1.1. Related Works

The authors of [8,9] applied Rician channels to MIMO systems. However, the study only concentrated on the one-hop scenario. Therefore, it can only cover a limited communication distance in real problems. In order to reduce the power consumption of ADCs, scholars have made plenty of attempts with the idea of using low-resolution ADCs. The authors of [10] studied the hybrid ADCs/DACs relay system and proposed the power scaling law. This law revealed that the transmission power could be reduced inversely proportional to the number of relay antennas, and an effective power allocation scheme was further proposed based on this law. The authors of [11] studied the one-bit low-resolution ADCs relay system for the MIMO system, which is a special case of low-resolution ADCs, and proved one-bit low-resolution ADCs is effective for reducing energy consumption. The authors of [12,13] applied low-resolution ADCs to the Rician channels relay system. However, they only considered the one-hop relay system scenario, so the communication distance was limited. Since it is very important for the MIMO system to increase the achievable sum rate, reduce the energy consumption and maintain long-distance communication, we consider the uplink of a two-hop low-resolution ADCs massive MIMO relaying system over the Rician channel in our paper.

### 1.2. Contributions

Our work considers the uplink of a two-hop low-resolution ADCs massive MIMO relay system over Rician channels. Firstly, we derive the closed-form expression of the achievable sum rate of the uplink of the low-resolution ADCs massive MIMO relay system over two-hop Rician channels and one-hop Rayleigh channels based on the higher-order statistics of perfect channel state information (CSI). Secondly, we derive the asymptotic closed-form expressions when the number of antennas tends to infinity. Next, we further achieve the law of power scaling and asymptotic values under different power scales, and conclude that the transmission power scaled down inversely proportional to the number of antennas at relays. Finally, we compare the achievable sum rates of the two-hop Rician system and the one-hop Rayleigh system, and validate the correctness of the theoretical analysis with numerical simulation results.

More specifically, the contributions of this work are summarised as follows:

1. We design a two-hop Rician channel system, which guarantees the LOS between users, relays and targets, and takes the advantage of Rician channels to increase the achievable sum rate while maintaining long-distance communication.

2. We use both mathematical approaches and simulation results to prove the superiority of achievable sum rate of our two-hop Rician channel system over the one-hop Rayleigh channel system, which is more widely applied to MIMO systems.

3. We apply low-resolution ADCs to our two-hop Rician system to reduce the energy consumption, and use mathematical analysis and simulation results to find that when quantization bits q = 4, the system achieves a fairly considerable achievable sum rate while greatly reduces the energy consumption.

### 1.3. Notations

Notation: The superscripts ${\left(\cdot \right)}^{T}$, ${\left(\cdot \right)}^{H}$, tr(·) and diag(·) represent the transpose, Hermitian transpose, trace of the matrix, and diagonal matrix, respectively. ||·|| represents the Euclidean norm. $CN(*,\sigma^{2})$ represents the complex Gaussian distribution with the mean of ∗ and the variance of $\sigma^{2}$. E[·] represents the expectation. $I_{N}$ denotes an N×N identify matrix. $X_{ij}$ or ${\left[X\right]}_{ij}$ represents the ${\left(i,j\right)}_{th}$ entry of X.

## 2. System Model and Signal Processing

### 2.1. One-Hop Rayleigh Channel System

Figure 1 shows the system model of the uplink of a one-hop massive MIMO relay with low-resolution ADCs under Rayleigh channels. This system contains relay $R_{1}$ with $N_{R1}$ antennas, and *K* users with single antenna. The system works under Rayleigh channels because the distances between users and the relay are very large and there is no LOS between users and the relay.

We use $G_{R1}\subset C^{N_{R1}\times K}$ to denote the MIMO channel matrix. According to [9], $G_{R1}$ can be represented as
(1)$$\begin{array}{cc} \; & G_{R1}\;=\;H_{R1}D_{R1}^{1/2} \end{array}$$
where $D_{R1}$ is a K×K diagonal matrix representing the large-scale fading between *K* users and *K* different randomly selected antennas from $N_{R1}$ antennas in relay $R_{1}$ with the probability of $1/N_{R1}$, and ${\left[D_{R1}\right]}_{kk}\;=\;\alpha_{k}$. $\alpha_{k}\;=\;{\left(\frac{d_{ref}}{d_{u\_R1}}\right)}^{v}$, where $d_{ref}$ represents the reference distance, $d_{i\_j}$ represents the distance from node *i* to node *j*, *v* is the power exponent coefficient.

$H_{R1}\subset C^{N_{R1}\times K}$ denotes the fast-fading matrix of the Rayleigh channels. Every column of $H_{R1}$ follows $CN(0,\frac{1}{2})$.

Assume that the signal transmitted by *K* user antennas is $x_{S}\;=\;{\left[x_{1},x_{2},\ldots ,x_{K}\right]}^{T}$, where $E\left[x_{S}x_{S}^{T}\right]\;=\;I_{K}$. After one time slot, the signal received by relay $R_{1}$ can be expressed as
(2)$$\begin{array}{cc} \; & y_{R1}\;=\;\sqrt{P_{u}}G_{R1}x_{S}\;+\;n_{R1} \end{array}$$
where $P_{u}$ is the transmission power of each user, $n_{R1}$ is the white noise follows i.i.d complex Gaussian distribution at relay $R_{1}$, $n_{R1}∼CN\left(0,\sigma_{R1}^{2}\right)$.

$y_{R1}$ is then quantized by low-resolution ADCs at $R_{1}$. Based on Additive Quantization Noise Model (AQNM) [14,15], the quantized signal can be represented as
(3)$$\begin{array}{cc} \; & {\tilde{y}}_{R1}\;=\;Q\left[y_{R1}\right]\;=\;my_{R1}\;+\;{\tilde{n}}_{R1} \end{array}$$
where ${\tilde{n}}_{R1}$ denotes the additive quantization noise vector and is independent from the received signal $y_{R1}$, *m* denotes the linear quantization gain. According to [10,16,17], *m* satisfies the following equation m = 1−ρ, where ρ represents the quantization distortion factor and equals the ratio of the quantizer error variance over received signal variance. For relay $R_{1}$, $\rho \;=\;\frac{E[|y_{R1}-{\tilde{y}}_{R1}{|}^{2}]}{E[|y_{R1}{|}^{2}]}$. When the number of quantization bits q≤5, the values of ρ is shown in Table 1. When q>5, $\rho \approx \frac{\pi \sqrt{3}}{2}\cdot 2^{-2q}$.

According to [18], the covariance matrix of the quantization noise can be expressed as
(4)$$\begin{array}{cc} \; & R_{{\tilde{n}}_{R1}}\;=\;m\rho diag\left(P_{u}G_{R1}G_{R1}^{H}\;+\;\sigma_{R1}^{2}I_{N_{R1}}\right) \end{array}$$

Because Maximum Ratio Combining (MRC) has low-complexity and is able to achieve the optimal reception performance, we use MRC to linear process the quantized signal ${\tilde{y}}_{R1}$, where the MRC matrix $W_{R1}^{H}\;=\;G_{R1}^{H}$. Therefore, the processed signal $x_{R1}$ can be written as
(5)$$\begin{array}{cc} \; & x_{R1}\;=\;W_{R1}^{H}{\tilde{y}}_{R1}\;=\;m\sqrt{P_{u}}G_{R1}^{H}G_{R1}x_{S}\;+\;mG_{R1}^{H}n_{R1}\;+\;G_{R1}^{H}{\tilde{n}}_{R1} \end{array}$$

Noticing that the signal of the $k_{th}$ user and the other users in (Equation 5) are uncorrelated, the received signal of the $k_{th}$ user at relay $R_{1}$ can be written as
(6)$$\begin{array}{cc} \; & x_{R1,k}\;=\;\underset{\mathrm{desired signal}}{\underbrace{m\sqrt{P_{u}}g_{R1,k}^{H}G_{R1}x_{S,k}}}\;+\;\underset{\mathrm{interference}}{\underbrace{m\sqrt{P_{u}}\Sigma_{j\neq k}^{K}g_{R1,j}^{H}G_{R1}x_{S,j}}}\;+\;\underset{\mathrm{noise}}{\underbrace{mg_{R1,k}^{H}n_{R1}\;+\;g_{R1,k}^{H}{\tilde{n}}_{R1}}} \end{array}$$

### 2.2. Two-Hop Rician Channel

In order to increase the achievable sum rate, we convert the one-hop MIMO system to two-hop MIMO system, so that the distance between users and relays can be reduced. As a result, LOS will appear between users and relays and Rician channels can be applied to increase the achievable sum rate.

Figure 2 shows the system model of the uplink of a two-hop massive MIMO system with low-resolution ADCs under Rician channels. This system contains relay $R_{1}$ with $N_{R1}$ antennas, relay $R_{2}$ with $N_{R2}$ antennas and *K* users with a single antenna. The system works under Rician channels because LOS exists between users and $R_{1}$ and between $R_{1}$ and $R_{2}$.

We use $G_{R1}\subset C^{N_{R1}\times K}$ to denote the MIMO channel matrix between users, $G_{R2}\subset C^{N_{R2}\times K}$ to denote the MIMO channel matrix between $R_{1}$ and $R_{2}$. $G_{R1}$ can be represented as
$$\begin{array}{cc} \; & G_{R1}\;=\;H_{R1}D_{R1}^{1/2} \end{array}$$
$G_{R2}$ can be represented as
(7)$$\begin{array}{cc} \; & G_{R2}\;=\;H_{R2}D_{R2}^{1/2} \end{array}$$
where $D_{R1}$ is a diagonal matrice representing the large-scale fading between *K* users and the *K* different randomly selected antennas from $N_{R1}$ antennas with the probability of $1/N_{R1}$ at relay $R_{1}$, and ${\left[D_{R1}\right]}_{kk}\;=\;\alpha_{k}$. $D_{R2}$ is a diagonal matrice representing the large-scale fading between *K* users and *K* different randomly selected antennas from $N_{R2}$ antennas at relay $R_{2}$ with the probability of $1/N_{R2}$, and ${\left[D_{R2}\right]}_{kk}\;=\;\beta_{k}$. $\alpha_{k}\;=\;{\left(\frac{d_{ref}}{d_{u\_R1}}\right)}^{v}$, $\beta_{k}\;=\;{\left(\frac{d_{ref}}{d_{R1\_R2}}\right)}^{v}$, where $d_{ref}$ represents the reference distance, $d_{i\_j}$ represents the distance from node *i* to node *j*, *v* is power exponent coefficient.

$H_{R1}\subset C^{N_{R1}\times K}$ and $H_{R2}\subset C^{N_{R2}\times K}$ denote the fast-fading matrix of Rician channels. According to [19], $H_{R1}$ and $H_{R2}$ can be represented as:(8)$$\begin{array}{cc} \; & H_{R1}\;=\;H_{R1}^{-}{\left[\Omega_{R1}{\left(\Omega_{R1}\;+\;I_{K}\right)}^{-1}\right]}^{1/2}\;+\;H_{R1}{\left[{\left(\Omega_{R2}\;+\;I_{K}\right)}^{-1}\right]}^{1/2} \end{array}$$
$$\begin{array}{cc} \; & G_{R2}\;=\;H_{R2}D_{R2}^{1/2} \end{array}$$

Same as the previous chapter, we can get the quantized signal ${\tilde{y}}_{R1}$ at $R_{1}$ as
$$\begin{array}{cc} \; & {\tilde{y}}_{R1}\;=\;Q\left[y_{R1}\right]\;=\;my_{R1}\;+\;{\tilde{n}}_{R1} \end{array}$$

From (Equation 4), we can get the covariance matrix of the quantization noise $R_{{\tilde{n}}_{R1}}$ at $R_{1}$ as
(9)$$\begin{array}{cc} \; & R_{{\tilde{n}}_{R1}}\;=\;m_{1}\rho_{1}diag\left(P_{u}G_{R1}G_{R1}^{H}\;+\;\sigma_{R1}^{2}I_{N_{R1}}\right) \end{array}$$
where $m_{1}$ denotes the linear quantization gain at $R_{1}$, $\rho_{1}$ denotes the quantization distortion factor at $R_{1}$.

The processed signal $x_{R1}$ can be expressed as
$$\begin{array}{cc} \; & x_{R1}\;=\;W_{R1}^{H}{\tilde{y}}_{R1}\;=\;m\sqrt{P_{u}}G_{R1}^{H}G_{R1}x_{S}\;+\;mG_{R1}^{H}n_{R1}\;+\;G_{R1}^{H}{\tilde{n}}_{R1} \end{array}$$

Then, we apply the technique of Amplify-and-Forward (AF) to signal $x_{R1}$ and transmit the processed signals to $R_{2}$ with *K* randomly selected antennas. The signal $y_{R2}$ received at relay $R_{2}$ can be denoted as
(10)$$\begin{array}{cc} \; & y_{R2}\;=\;\gamma G_{R2}x_{R1}\;+\;n_{R1} \end{array}$$
where $G_{R2}$ is the Rayleigh fading channel between relay $R_{1}$ and $R_{2}$, $n_{R2}∼CN\left(0,\sigma_{R2}^{2}\right)$, which is the white noise follows i.i.d complex Gaussian distribution at relay $R_{2}$. γ is an amplification factor at relay $R_{1}$, which satisfies the power constraint $E[∥\gamma x_{R1}{∥}^{2}]\;=\;P_{R}$. Therefore, γ can be expressed as
(11)$$\begin{array}{cc} \; & \gamma \;=\;\sqrt{\frac{p_{R}}{E[∥x_{R1}{∥}^{2}]}} \end{array}$$
where $P_{R}$ represents the transmit power at relay $R_{1}$,

$E[∥x_{R1}{∥}^{2}]\;=\;m_{1}^{2}\left[p_{u}tr\left(E\left[G_{R1}^{H}G_{R1}G_{R1}^{H}G_{R1}\right]\right)\;+\;\sigma_{R1}^{2}tr\left(E\left[G_{R1}^{H}G_{R1}\right]\right)\right]\;+\;tr\left(E\left[G_{R1}^{H}R_{{\tilde{n}}_{R1}}G_{R1}\right]\right)$.

To simplify the expression, we make the following definitions.
$$\Delta_{R1,k}\;=\;\frac{2\mu_{k}\;+\;1}{{\left(\mu_{k}\;+\;1\right)}^{2}}\;\;\;\;\Phi_{R1,ki}\;=\;\frac{sin\left(N_{R1}\pi \left(sin\theta_{R1}-sin\theta_{R1,i}\right)/2\right)}{sin\left(\pi (sin\theta_{R1,k}-sin\theta_{R1,i})/2\right)}\;\;\;\;Q_{ki}\;=\;\frac{\frac{\mu_{k}\mu_{i}\Phi_{ki}^{2}}{N_{R1}}\;+\;\mu_{k}\;+\;\mu_{i}\;+\;1}{\left(\mu_{k}\;+\;1\right)\left(\mu_{i}\;+\;1\right)}$$
$$\Delta_{R2,k}\;=\;\frac{2\varepsilon_{k}\;+\;1}{{\left(\varepsilon_{k}\;+\;1\right)}^{2}}\;\;\;\;\Phi_{R2,ki}\;=\;\frac{sin\left(N_{R2}\pi \left(sin\theta_{R2}-sin\theta_{R2,i}\right)/2\right)}{sin\left(\pi (sin\theta_{R2,k}-sin\theta_{R2,i})/2\right)}\;\;\;\;R_{ki}\;=\;\frac{\frac{\varepsilon_{k}\varepsilon_{i}\Phi_{ki}^{2}}{N_{R2}}\;+\;\varepsilon_{k}\;+\;\varepsilon_{i}\;+\;1}{\left(\varepsilon_{k}\;+\;1\right)\left(\varepsilon_{i}\;+\;1\right)}$$

Therefore, γ can be expressed as follows, the proof is attached in Appendix A.
(12)$$\begin{array}{cc} \; & \gamma \;=\;\sqrt{\frac{P_{R}}{m_{1}^{2}\left(P_{u}S_{1}\;+\;N_{R1}\sigma_{R1}^{2}\Sigma_{i\;=\;1}^{K}\alpha_{i}\right)\;+\;m_{1}\rho_{1}S_{2}}}\\ \; & S_{1}\;=\;N_{R1}\Sigma_{i\;=\;1}^{K}\alpha_{i}^{2}\left(N_{R1}\;+\;\Delta_{R1,i}\right)\;+\;N_{R1}\Sigma_{i\;=\;1}^{K}\alpha_{i}\Sigma_{l\;=\;1}^{K}\alpha_{l}Q_{il}\\ \; & S_{2}\;=\;p_{u}N_{R1}\Sigma_{n\;=\;1}^{K}\alpha_{n}\left(\alpha_{n}\;+\;\Sigma_{i\;=\;1}^{K}\alpha_{i}\right)\;+\;N_{R1}\sigma_{R1}^{2}\Sigma_{n\;=\;1}^{K}\alpha_{n} \end{array}$$

Similar to the quantization at relay $R_{1}$, the quantized signal ${\tilde{y}}_{R2}$ at relay $R_{2}$ can be modeled as
(13)$$\begin{array}{cc} \; & {\tilde{y}}_{R2}\;=\;Q\left[y_{R2}\right]\;=\;m_{1}y_{R2}\;+\;{\tilde{n}}_{R2} \end{array}$$

The covariance matrix of the quantization noise ${\tilde{n}}_{R1}$ can be written as
(14)$$\begin{array}{cc} \; & R_{{\tilde{n}}_{R2}}\;=\;m_{2}\rho_{2}diag\left(\gamma^{2}R_{y_{R2}}\;+\;\sigma_{R2}^{2}I_{N_{R2}}\right) \end{array}$$
where $m_{2}$ denotes the linear quantization gain at $R_{2}$, $\rho_{2}$ denotes the quantization distortion factor at $R_{2}$, $R_{y_{R2}}\;=\;G_{R2}G_{R1}^{H}R_{y_{R1}}G_{R1}G_{R2}^{H}$, $R_{{\tilde{y}}_{R2}}\;=\;m_{1}^{2}\left(P_{u}G_{R1}G_{R1}^{H}\;+\;\sigma_{R1}^{2}I_{N_{R1}}\right)\;+\;m_{1}\rho_{1}diag\left(P_{u}G_{R1}G_{R1}^{H}\;+\;\sigma_{R1}^{2}I_{N_{R1}}\right)$.

Same as MRC processing at relay $R_{1}$, we also use MRC to process signals at $R_{2}$, where the MRC matrix $W_{R2}^{H}\;=\;G_{R2}^{H}$. Therefore, the processed signal $x_{R2}$ can be written as
(15)$$\begin{array}{cc} \; & x_{R2}\;=\;W_{R2}^{H}{\tilde{y}}_{R2}\;=\;\gamma m_{1}m_{2}\sqrt{P_{u}}G_{R2}^{H}G_{R2}G_{R1}^{H}G_{R1}x_{S}\;+\;\gamma m_{1}m_{2}G_{R2}^{H}G_{R2}G_{R1}^{H}n_{R1}\\ \; & \;\;\;\;\;\;\;\;\;\;\;\;\;\;\;\;\;\;\;\;\;\;\;\;\;\;\;\;\;\;\;\;\;\;\;\;\;+\;\gamma m_{2}G_{R2}^{H}G_{R2}G_{R1}^{H}{\tilde{n}}_{R1}\;+\;m_{2}G_{R2}^{H}n_{R2}\;+\;G_{R2}^{H}{\tilde{n}}_{R1} \end{array}$$

Noticing that the signal of the $k_{th}$ user and the other users in (Equation 15) are uncorrelated, the received signal of the $k_{th}$ user at relay $R_{2}$ can be written as
(16)$$\begin{array}{cc} \; & x_{R2,k}\;=\;\underset{\mathrm{desired signal}}{\underbrace{\gamma m_{1}m_{2}\sqrt{P_{u}}g_{R2,k}^{H}G_{R2}G_{R1}^{H}g_{R1,k}x_{S,k}}}\;+\;\underset{\mathrm{interference}}{\underbrace{\gamma m_{1}m_{2}\sqrt{P_{u}}\Sigma_{j\neq k}^{K}g_{R2,k}^{H}G_{R2}G_{R1}^{H}g_{R1,j}x_{S,j}}}\\ \; & \;\;\;\;\;\;\;\;\;\;\;\;\;\;\;\;\;+\;\underset{\mathrm{noise}}{\underbrace{\gamma m_{1}m_{2}g_{R2,k}^{H}G_{R2}G_{R1}^{H}n_{R1}\;+\;\gamma m_{2}g_{R2,k}^{H}G_{R2}G_{R1}^{H}{\tilde{n}}_{R1}\;+\;m_{2}g_{R2,k}^{H}n_{R2}\;+\;g_{R2,k}^{H}{\tilde{n}}_{R1}}} \end{array}$$

## 3. System Achievable Rate Analysis

### 3.1. One-Hop Rayleigh Channel

#### 3.1.1. Closed-Form Expression

Supposing that the CSI is perfect, based on Shannon Entropy and according to (Equation 6), we can get the rate of the $k_{th}$ user in one-hop low-precision ADCs MIMO relay system over Rayleigh channels as
(17)$$\begin{array}{cc} \; & R_{k}^{Rayleigh}\;=\;\frac{1}{2}E\left[log_{2}\left(1\;+\;\frac{P_{k}^{Rayleigh}}{N_{k}^{Rayleigh}}\right)\right] \end{array}$$
(18)$$\begin{array}{cc} \; & P_{k}^{Rayleigh}\;=\;m^{2}P_{u}{\left|g_{R1,k}^{H}G_{R1}\right|}^{2} \end{array}$$
(19)$$\begin{array}{cc} \; & N_{k}^{Rayleigh}\;=\;m^{2}P_{u}\Sigma_{j\neq k}^{K}|g_{R1,j}^{H}G_{R1}{|}^{2}\;+\;m^{2}\sigma_{R1}^{2}|g_{R1,k}^{H}{|}^{2}\;+\;\left|g_{R1,k}^{H}R_{{\tilde{n}}_{R1}}g_{R1,k}\right| \end{array}$$
where $P_{k}^{Rayleigh}$ represents the power of desired signal of the $k_{th}$ user, $N_{k}^{Rayleigh}$ represents the power of interference signal and the noise of the $k_{th}$ user.

According to [20], the rate of $k_{th}$ user can also be denoted as
(20)$$\begin{array}{cc} \; & R_{k}^{Rayleigh}\;=\;\frac{1}{2}log_{2}\left(1\;+\;SNR_{k}\right) \end{array}$$
where $SNR_{k}$ represents the Signal-to-noise Ratio (SNR) of the $k_{th}$ user at the receiving end $R_{1}$.
(21)$$\begin{array}{cc} \; & SNR_{k}\;=\;\frac{P_{k}^{Rayleigh^{\prime }}}{N_{k}^{Rayleigh^{\prime }}} \end{array}$$

Based of (Equation 20) and (Equation 21), we can derive the closed-form expression for the achievable sum rate of the $k_{th}$ user in the one-hop low-precision ADCs MIMO relay system over Rayleigh channels is
(22)$$\begin{array}{cc} \; & R_{k}^{Rayleigh}\;=\;\frac{1}{2}log_{2}\left(1\;+\;\frac{P_{k}^{Rayleigh^{\prime }}}{N_{k}^{Rayleigh^{\prime }}}\right) \end{array}$$

In Formula (Equation 22), $P_{k}^{Rayleigh^{\prime }}$ and $N_{k}^{Rayleigh^{\prime }}$ can be represented as follows, the proof is attached in Appendix B.
(23)$$\begin{array}{cc} \; & P_{k}^{Rayleigh^{\prime }}\;=\;m^{2}P_{u}E[|g_{R1,k}^{H}G_{R1}{|}^{2}]\\ \; & \;\;\;\;\;\;\;\;\;\;\;\;\;\;\;\;\;\;\;\;\;=\;m^{2}P_{u}\alpha_{k}^{2}N_{R1}\left(N_{R1}\;+\;1\right) \end{array}$$
(24)$$\begin{array}{cc} \; & N_{k}^{Rayleigh^{\prime }}\;=\;m^{2}P_{u}\Sigma_{j\neq k}^{K}E[|g_{R1,j}^{H}G_{R1}{|}^{2}]\\ \; & \;\;\;\;\;\;\;\;\;\;\;\;\;\;\;\;\;\;\;\;\;\;\;\;\;\;+\;m^{2}\sigma_{R1}^{2}E[|g_{R1,k}^{H}{|}^{2}]\;+\;E[|g_{R1,k}^{H}R_{{\tilde{n}}_{R1}}g_{R1,k}\left|\right]\\ \; & \;\;\;\;\;\;\;\;\;\;\;\;\;\;\;\;\;\;\;\;\;=\;m^{2}P_{u}N_{R1}\Sigma_{j\neq k}^{K}\alpha_{k}\alpha_{j}\;+\;m^{2}\sigma_{R1}^{2}N_{R1}\alpha_{k}\\ \; & \;\;\;\;\;\;\;\;\;\;\;\;\;\;\;\;\;\;\;\;\;\;\;\;\;\;+\;m\rho \alpha_{n}N_{R1}\left(P_{u}\Sigma_{i\;=\;1}^{K}\alpha_{i}\;+\;P_{u}\alpha_{n}\;+\;\sigma_{R1}^{2}\right) \end{array}$$

#### 3.1.2. Power Scaling Laws and Asymptotic Analysis

Based on the closed-form expression over Rayleigh channels given by (Equation 22), we further analyze their performances and derive the law of energy scaling in different conditions.

Suppose that the transmit power at the user end is $P_{u}\;=\;\frac{E_{u}}{N_{R1}^{a}}$, *a* is the power scaling constant. When the number of antennas tends to infinity, the limit of $SNR_{k}$ in (Equation 21) can be represented as
(25)$$\begin{array}{cc} \; & \lim_{N_{R1}\to \infty }SNR_{k}\;=\;\lim_{N_{R1}\to \infty }\frac{\alpha_{k}^{2}m^{2}E_{u}N_{R1}^{2-a}}{\alpha_{k}m^{2}\sigma_{R1}^{2}N_{R1}\;+\;\Sigma_{n\;=\;1}^{K}\alpha_{n}m\rho \sigma_{R1}^{2}N_{R1}}\;=\;\lim_{N_{R1}\to \infty }\frac{\alpha_{k}^{2}mE_{u}N_{R1}^{1-a}}{\alpha_{k}m\sigma_{R1}^{2}\;+\;\Sigma_{n\;=\;1}^{K}\alpha_{n}\rho \sigma_{R1}^{2}} \end{array}$$

When *a* takes different values, we can obtain the following power scaling law
(26)$$\begin{array}{cc} \; & \lim_{N_{R1}\to \infty }SNR_{k}\;\;=\;\left\{\begin{array}{cc} \infty & a<1\\ \frac{\alpha_{k}^{2}mE_{u}}{\alpha_{k}m\sigma_{R1}^{2}\;+\;\Sigma_{n\;=\;1}^{K}\alpha_{n}\rho \sigma_{R1}^{2}} & a\;=\;1\\ 0 & a\geq 1 \end{array}\right. \end{array}$$

As can be seen from (Equation 26), when $N_{R1}$ tends to infinity, the transmit power at the user end can be scaled down by $\frac{1}{N_{R1}^{a}}$. When the scale index *a* satisfies a = 1, the achievable rate remains stable. Based on (Equation 21), (Equation 22) and (Equation 26), when the prefect CSI exists from users to $R_{1}$, we assume that $N_{R1}\gg K\gg 1$, the large-scale fading between users and R1 satisfies $\alpha_{1}\;=\;\alpha_{2}\;=\;\cdots \;=\;\alpha_{k}\;=\;\alpha$, the user transmit power $P_{u}\gg \sigma_{R1}^{2}$, $SNR_{k}$ can be approximated as
(27)$$\begin{array}{cc} \; & SNR_{k}\approx \frac{mN_{R1}}{K} \end{array}$$

The proof is attached in Appendix C.

The uplink achievable sum rate can be approximated as
(28)$$\begin{array}{cc} \; & R_{sum}^{Rayleigh}\;=\;\frac{K}{2}log_{2}\left(1\;+\;\frac{mN_{R1}}{K}\right) \end{array}$$

### 3.2. Two-Hop Rician Channels

#### 3.2.1. Closed-Form Expression

Similar to (Equation 17), the achievable rate of the $k_{th}$ user in two-hop low-precision ADCs MIMO relay system over Rician channels can be represented as
(29)$$\begin{array}{cc} \; & R_{k}^{Rician}\;=\;\frac{1}{2}E\left[log_{2}\left(1\;+\;\frac{P_{k}^{Rician}}{N_{k}^{Rician}}\right)\right] \end{array}$$
where $P_{k}^{Rician}$ represents the power of desired signal of the $k_{th}$ user, and $N_{k}^{Rician}$ represents the power of interference signal and the of noise of the $k_{th}$ user.

The detailed formula of $P_{k}^{Rician}$ and $N_{k}^{Rician}$ can also be represented as
(30)$$\begin{array}{cc} \; & P_{k}^{Rician}\;=\;\gamma^{2}m_{1}^{2}m_{2}^{2}P_{u}{\left|g_{R2,k}^{H}G_{R2}G_{R1}^{H}g_{R1,k}\right|}^{2} \end{array}$$
(31)$$\begin{array}{cc} \; & N_{k}^{Rician}\;=\;\gamma^{2}m_{1}^{2}m_{2}^{2}P_{u}\Sigma_{j\neq k}^{K}|g_{R2,k}^{H}G_{R2}G_{R1}^{H}g_{R1,j}{|}^{2}\;+\;\gamma^{2}m_{1}^{2}m_{2}^{2}\sigma_{R1}^{2}{\left|g_{R2,k}^{H}G_{R2}G_{R1}^{H}\right|}^{2}\\ \; & \;\;\;\;\;\;\;\;\;\;\;\;\;\;\;\;\;\;\;\;\;+\;\gamma^{2}m_{2}^{2}|g_{R2,k}^{H}G_{R2}G_{R1}^{H}R_{{\tilde{n}}_{R1}}G_{R1}G_{R2}^{T}g_{R2,k}{|}^{2}\;+\;m_{2}^{2}\sigma_{R2}^{2}{\left|g_{R2,k}\right|}^{2}\\ \; & \;\;\;\;\;\;\;\;\;\;\;\;\;\;\;\;\;\;\;\;\;+\;|g_{R2,k}^{H}R_{{\tilde{n}}_{R2}}g_{R2,k}{|}^{2} \end{array}$$

Similar to (Equation 20), the rate of $k_{th}$ user can be denoted as
(32)$$\begin{array}{cc} \; & R_{k}^{Rician}\;=\;\frac{1}{2}log_{2}\left(1\;+\;SNR_{k}\right) \end{array}$$
where $SNR_{k}$ represents the SNR of the $k_{th}$ user at the receiving end $R_{2}$.
(33)$$\begin{array}{cc} \; & SNR_{k}\;=\;\frac{P_{k}^{Rician^{\prime }}}{N_{k}^{Rician^{\prime }}} \end{array}$$

Based of (Equation 32) and (Equation 33), we can derive the closed-form expression for the achievable sum rate of the two-hop low-precision ADCs MIMO relay system over Rician channels is
(34)$$\begin{array}{cc} \; & R_{k}^{Rician}\;=\;\frac{1}{2}log_{2}\left(1\;+\;\frac{P_{k}^{Rician^{\prime }}}{N_{k}^{Rician^{\prime }}}\right) \end{array}$$

In Formula (Equation 34), $P_{k}^{Rayleigh^{\prime }}$ and $N_{k}^{Rayleigh^{\prime }}$ can be represented as follows, the proof is attached in Appendix D.
(35)$$\begin{array}{cc} \; & P_{k}^{Rician^{\prime }}\;=\;\gamma^{2}m_{1}^{2}m_{2}^{2}P_{u}E[|g_{R2,k}^{H}G_{R2}G_{R1}^{H}g_{R1,k}{|}^{2}]\\ \; & \;=\;\gamma^{2}m_{1}^{2}m_{2}^{2}P_{u}\alpha_{k}\beta_{k}N_{R1}N_{R2}\left[\alpha_{k}\beta_{k}\left(N_{R1}\;+\;\Delta_{R1,k}\right)\left(N_{R2}\;+\;\Delta_{R2,k}\right)\;+\;\Sigma_{i\neq k}^{K}\alpha_{i}\beta_{i}Q_{ki},R_{ki}\right] \end{array}$$
(36)$$\begin{array}{cc} \; & N_{k}^{Rician^{\prime }}\;=\;A_{k}^{Rician}\;+\;B_{k}^{Rician}\;+\;C_{k}^{Rician}\;+\;D_{k}^{Rician}\;+\;E_{k}^{Rician}\\ \; & A_{k}^{Rician}\;=\;\gamma^{2}m_{1}^{2}m_{2}^{2}P_{u}\Sigma_{j\neq k}^{K}E[|g_{R2,k}^{H}G_{R2}G_{R1}^{H}g_{R1,j}{|}^{2}]\\ \; & \;\;\;\;\;\;\;\;\;\;\;\;\;\;\;\;=\;\gamma^{2}m_{1}^{2}m_{2}^{2}P_{u}\beta_{k}N_{R1}N_{R2}\Sigma_{j\neq k}^{K}\alpha_{j}[\alpha_{k}\beta_{k}Q_{kj}\left(N_{R2}\;+\;\Delta_{R2,k}\right)\\ \; & \;\;\;\;\;\;\;\;\;\;\;\;\;\;\;\;\;\;\;\;\;+\;\alpha_{j}\beta_{j}R_{kj}\left(N_{R1}\;+\;\Delta_{R1,j}\right)\;+\;\Sigma_{i\neq k}^{K}\alpha_{i}\beta_{i}R_{ki}Q_{ij}]\\ \; & B_{k}^{Rician}\;=\;\gamma^{2}m_{1}^{2}m_{2}^{2}\sigma_{R1}^{2}E[|g_{R2,k}^{H}G_{R2}G_{R1}^{H}{|}^{2}]\\ \; & \;\;\;\;\;\;\;\;\;\;\;\;\;\;\;\;=\;\gamma^{2}m_{1}^{2}m_{2}^{2}\sigma_{R1}^{2}\beta_{k}N_{R1}N_{R2}\left[\alpha_{k}\beta_{k}\left(N_{R2}\;+\;\Delta_{R2,k}\right)\;+\;\Sigma_{i\neq k}^{K}\alpha_{i}\beta_{j}R_{ki}\right]\\ \; & C_{k}^{Rician}\;=\;\gamma^{2}m_{2}^{2}E\left[g_{R2,k}^{H}G_{R2}G_{R1}^{H}R_{{\tilde{n}}_{R1}}G_{R1}G_{R2}^{H}g_{R2,k}\right]\\ \; & \;\;\;\;\;\;\;\;\;\;\;\;\;\;\;\;=\;\gamma^{2}m_{1}\rho_{1}m_{2}^{2}\beta_{k}N_{R1}N_{R2}[P_{u}(\left(N_{R2}\;+\;\Delta_{R2,k}\right)\alpha_{k}\beta_{k}\left(\alpha_{k}\;+\;\Sigma_{l\neq i}^{K}\alpha_{l}\right)\\ \; & \;\;\;\;\;\;\;\;\;\;\;\;\;\;\;\;\;\;\;\;\;+\;\Sigma_{i\neq k}^{K}\alpha_{i}\beta_{i}R_{ki}\left(\alpha_{i}\;+\;\Sigma_{l\neq i}^{K}\alpha_{l}\right))\;+\;\sigma_{R1}^{2}\left(\alpha_{k}\beta_{k}\left(N_{R2}\;+\;\Delta_{R2,k}\right)\;+\;\Sigma_{i\neq k}^{K}\alpha_{i}\beta_{i}R_{ki}\right)]\\ \; & D_{k}^{Rician}\;=\;m_{2}^{2}\sigma_{R2}^{2}E[|g_{R2,k}{|}^{2}]\;=\;m_{2}^{2}\sigma_{R2}^{2}N_{R2}\beta_{k}\\ \; & E_{k}^{Rician}\;=\;E\left[g_{R2,k}^{H}R_{{\tilde{n}}_{R1}}g_{R2,k}\right]\\ \; & \;\;\;\;\;\;\;\;\;\;\;\;\;\;\;\;=\;\gamma^{2}m_{1}m_{2}\rho_{2}N_{R1}N_{R2}\beta_{k}(\alpha_{k}\beta_{k}\Delta_{R2,k}\left(P_{u}\Sigma_{i\;=\;1}^{K}\alpha_{i}\;+\;m_{1}N_{R1}P_{u}\alpha_{k}\;+\;\rho_{1}P_{u}\alpha_{k}\;+\;\sigma_{R1}^{2}\right)\\ \; & \;\;\;\;\;\;\;\;\;\;\;\;\;\;\;\;\;\;\;\;\;+\;\Sigma_{n\;=\;1}^{K}\alpha_{n}\beta_{n}\left(p_{u}\Sigma_{i\;=\;1}^{K}\alpha_{i}\;+\;m_{1}N_{R1}P_{u}\alpha_{n}\;+\;\rho_{1}p_{u}\alpha_{n}\;+\;\sigma_{R1}^{2}\right))\;+\;m_{2}\rho_{2}\sigma_{R2}^{2}N_{R2}\beta_{k} \end{array}$$

#### 3.2.2. Power Scaling Laws and Asymptotic Analysis

Based on the closed-form expressions over Rician channel given by (Equation 33) and (Equation 34), we further analyze their performances and derive the law of energy scaling in different conditions.

Suppose that the transmit power at the user end is $P_{u}\;=\;\frac{E_{u}}{N_{R1}^{a}}$, the transmit power at relay $R_{1}$ is $P_{R}\;=\;\frac{E_{u}}{N_{R2}^{b}}$, *a* and *b* are power scaling constants, $\lambda \;=\;\frac{N_{R2}}{N_{R1}}<\infty$. When the $N_{R1}$ and $N_{R2}$ tend to infinity, the limit of $SNR_{k}$ in (Equation 33) can be represented as
(37)$$\begin{array}{cc} \; & \lim_{N_{R1}\to \infty }SNR_{k}\;=\;\lim_{N_{R1}\to \infty }\frac{m_{1}^{2}m_{2}\gamma^{2}P_{u}N_{R1}^{2}N_{R2}\alpha_{k}^{2}\beta_{k}}{m_{1}m_{2}\gamma^{2}N_{R1}N_{R2}\alpha_{k}\beta_{k}\sigma_{R1}^{2}\;+\;\sigma_{R2}^{2}}\\ \; & \;\;\;\;\;\;\;\;\;\;\;\;\;\;\;\;\;\;\;\;\;\;\;\;\;\;\;\;\;\;=\;\lim_{N_{R1}\to \infty }\frac{m_{1}^{2}m_{2}\lambda E_{u}\alpha_{k}N_{R1}^{1-a}}{m_{1}m_{2}\lambda \sigma_{R1}^{2}\;+\;\frac{\sigma_{R2}^{2}}{\gamma^{2}\alpha_{k}\beta_{k}N_{R1}^{2}}} \end{array}$$

When *a* and *b* take different values, we can obtain the following power scaling law, the proof is attached in Appendix E.
(38)$$\begin{array}{cc} \; & \lim_{N_{R1}\to \infty }SNR_{k}\;=\;\;\left\{\begin{array}{cc} \infty & a,b<1\\ \frac{m_{2}E_{R}\alpha_{k}^{2}\beta_{k}}{\sigma_{R2}^{2}\Sigma_{i\;=\;1}^{K}\alpha_{i}^{2}} & a<b\;=\;1\\ \frac{m_{1}E_{u}\alpha_{k}}{\sigma_{R1}^{2}} & b<a\;=\;1\\ \frac{m_{1}m_{2}E_{U}E_{R}\alpha_{k}^{2}\beta_{k}}{\tau } & a\;=\;b\;=\;1\\ 0 & a>1\;or\;b>1 \end{array}\right. \end{array}$$
where $\tau \;=\;m_{2}E_{R}\alpha_{k}\beta_{k}\sigma_{R1}^{2}\;+\;\sigma_{R2}^{2}\left(m_{1}E_{u}\Sigma_{i\;=\;1}^{K}\alpha_{i}^{2}\;+\;\sigma_{R1}^{2}\Sigma_{i\;=\;1}^{K}\alpha_{i}\right)$. As can be seen from (Equation 38), when $N_{R1}$ tends to infinity, the transmit power at the user end can be scaled down by $\frac{1}{N_{R1}^{a}}$ and $\frac{1}{N_{R2}^{b}}$. When the scale index *a* and *b* satisfy a<b = 1, b<a = 1 or a = b = 1, the achievable rate remains stable.

Based on (Equation 38) and according to [13], when the prefect CSI exists from users to $R_{1}$ and from $R_{1}$ to $R_{2}$, we assume that $N_{R2}>N_{R1}\gg K\gg 1$, the large-scale fading between users and R1 satisfies $\alpha_{1}\;=\;\alpha_{2}\;=\;\cdots \;=\;\alpha_{k}\;=\;\alpha$, the fading between $R_{1}$ and $R_{2}$ satisfies $\beta_{1}\;=\;\beta_{2}\;=\;\cdots \;=\;\beta_{k}\;=\;\beta$, the user transmit power $p_{u}\gg \sigma_{R1}^{2}$, $R_{1}$ transmit power $p_{R}\gg \sigma_{R2}^{2}$, $\lambda \;=\;\frac{N_{R2}}{N_{R1}}<\infty$, $SNR_{k}$ can be approximated as
(39)$$\begin{array}{cc} \; & SNR_{k}\approx \frac{m_{1}N_{R1}}{K} \end{array}$$

Therefore, no matter the system is under Rayleigh channel or Rician channel, we can derive the approximate uplink achievable sum rate as
(40)$$\begin{array}{cc} \; & R_{sum}^{Rician}\;=\;\frac{K}{2}log2\left(1\;+\;\frac{m_{1}N_{R1}}{K}\right) \end{array}$$

The proof is attached in Appendix F.

## 4. Results

In this section, we use system 1 to represent the one-hop Rayleigh system, system 2 to represent the two-hop Rician system. We set different experiments and visualize the 1000-time Monte Carlo simulation results and the achievable sum rate calculated from the closed-form expressions. Then we compare the results of system 1 and system 2 to verify the correctness of theoretical analysis. In our experiments, we set the number of users K = 10, the transmission power of users $P_{u}\;=\;20dB$, the transmission energy $P_{R}\;=\;25dB$, the noise energy at $R_{1}$ and $R_{2}$ as $\sigma_{R1}^{2}\;=\;1dB$, $\sigma_{R2}^{2}\;=\;1dB$ respectively. We assume $N_{R2}\;=\;4N_{R1}$, the large-scale fading coefficient $\alpha_{k}\;=\;{\left(\frac{d_{ref}}{d_{u\_R1}}\right)}^{v}$, $\beta_{k}\;=\;{\left(\frac{d_{ref}}{d_{R1\_R2}}\right)}^{v}$, where $d_{ref}$ represents the reference distance, $d_{i\_j}$ represents the distance from node *i* to node *j*, *v* is the power exponent coefficient. During the simulation, we set $d_{ref}\;=\;100m$, $d_{R1\_R2}\;=\;150m$, v = 2.4. In system 1, we set $d_{u\_R1}\;=\;\left[700,1136,1096,694,285,872,531,489,440,356\right]m$. In system 2, we set $d_{R1\_R2}\;=\;\left[550,986,946,544,135,722,381,339,290,206\right]m$. We use $\theta_{R1,i}$ to represent the arrival angle from users to $R_{1}$, $\theta_{R2,j}$ to represent the arrival angle from $R_{1}$ to $R_{2}$, $\theta_{R2,i}$ and $\theta_{R2,j}$ obeys the uniform distribution on $\left[-\frac{\pi }{2},\frac{\pi }{2}\right]$.

### 4.1. Experiment 1: Achievable Sum Rates with Different $N_{R1}$

Figure 3 shows the variation curve of the achievable sum rate in systems 1 and 2 with the variation of $N_{R1}$. The asterisks indicate the experimental result obtained through 1000 times Monte Carlo simulations, and the circles indicate the simulation result of the achievable sum rate calculated by the closed-form expression (Equation 22) and (Equation 34). As is shown in Figure 3, the curve of the Monte Carlo simulations perfectly matches the curve derived from the closed-form expressions, which proves the correctness of the derived closed-form expressions (Equation 22) and (Equation 34). Obviously, when the simulation parameters are the same, the achievable sum rate of the two-hop Rican system is higher than the achievable sum rate of the one-hop Rayleigh system. This result proves that converting one-hop Rayleigh system to two-hop Rician system can improve the achievable sum rate. It is in consistent with the actual communication process where there is LOS signal, and the communication quality under Rician channel is better.

### 4.2. Experiment 2: Achievable Sum Rates with Different q

Figure 4 shows the variation curve of the achievable sum rate in systems 1 and 2 with the variation of *q* when $N_{R1}\;=\;200,\;400,\;800$. Apart from previous findings, we can also discover that the low-resolution quantization brings performance loss. This is because when the quantization occurs, it reduces the SNR and causes performance degradation. We can also discover that when the number of quantization bits q≥4, the achievable sum rate can persist at a stable rate.

### 4.3. Experiment 3: ADC Energy Efficiency with Different q

Figure 5 shows the variation curve of the ADC Energy Efficiency in systems 1 and 2 with the variation of *q* when $N_{R1}\;=\;200,\;400,\;800$. In Figure 5, the ADC energy efficiency can be obtained by $EE\;=\;\frac{R}{P}$, where *R* represents the achievable sum rate, *P* represents the energy loss. According to [20,21], $P\;=\;c_{0}N_{R1}*2^{q}\;+\;c_{1}$, $c_{0}\;=\;0.0001W,c_{1}\;=\;0.02W$. The result shows that in both systems, the energy efficiency of the ADC shows a logarithmic downtrend when *q* increases, which denotes that the low-resolution ADC can improve the energy efficiency and reduce the energy consumption during signal transmission. Besides, we can clearly find that when *q* is small, system 2 has a higher ADC energy efficiency compared with system 1, which proves the superiority of system 2.

### 4.4. Experiment 4: Asymptotic Achievable Sum Rates in Two Systems

Figure 6 shows the variation curve of achievable sum rates when $N_{R1}$ increases with different scaling indexes and the corresponding asymptotic values. When $N_{R1}$ is relatively small, the results of the 1000-time Monte Carlo simulation do not match with the analytical results precisely. However, as the number of antennas continues to increase, the 1000-time Monte Carlo simulation results can perfectly match the analytical results. It is because the law of power scaling is derived when $N_{R1}$ is large enough.

Besides, Figure 6 shows that in system 1, $P_{u}$ can be scaled down inversely proportional to $N_{R1}$ when scaling index a = 1 while maintain a desirable achievable sum rate when $N_{R1}$ grows large. In system 2, $P_{u}$ and $P_{R}$ can be scaled down inversely proportional to $N_{R1}$ and $N_{R2}$ when a = 0,b = 1, or a = 1,b = 0, or a = 1,b = 1 and maintain desirable achievable sum rates when $N_{R1}$ grows large. The results shown in Figure 6 are consistent with the theoretical analysis given by Equations (Equation 26) and (Equation 38).

## 5. Conclusions

In this paper, we investigate the uplink of a two-hop low-resolution ADCs massive MIMO relaying system over Rician channels and compare its superiority of the achievable sum rate with the one-hop Rayleigh channel system. Firstly, we use the higher-order statistics to derive the closed-form expression of achievable sum rate. From the simulation results, we discover that converting a one-hop Rayleigh channel system into a two-hop Rician channel system can increase the achievable sum rate. Besides, the use of low-resolution ADCs only causes limited loss of achievable sum rate, but greatly improves the energy efficiency. Secondly, we discover that as the number of relay antennas continues to increase, the achievable sum rate eventually reaches a stable state. Finally, the power scaling law shows that when the number of antennas at the relay grows large, both $P_{u}$ and $P_{R}$ can be scaled down inversely proportional to $N_{R1}$ and $N_{R2}$, while maintaining a desirable achievable sum rate.

## Figures and Tables

**Figure 1 entropy-23-01074-f001:**
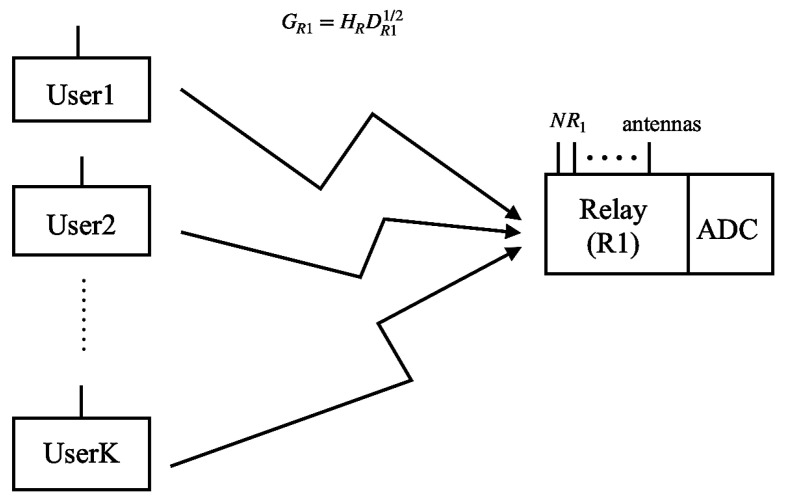
System model of a one-hop MIMO system under Rayleigh channels.

**Figure 2 entropy-23-01074-f002:**
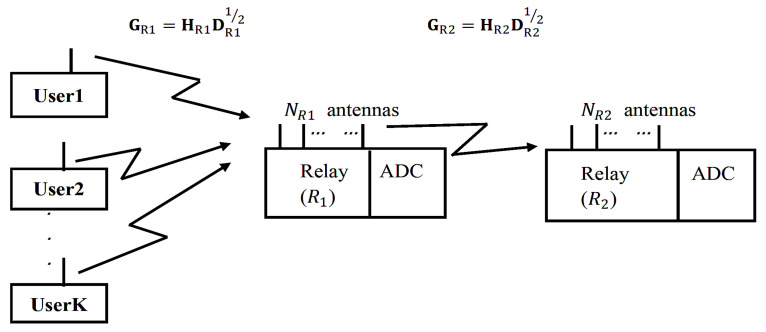
System model of a two-hop MIMO system under Rician channels.

**Figure 3 entropy-23-01074-f003:**
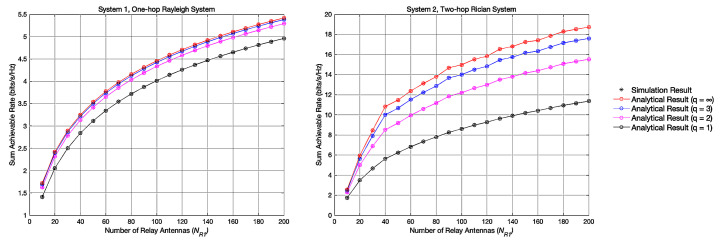
Achievable sum rate with different $N_{R1}$.

**Figure 4 entropy-23-01074-f004:**
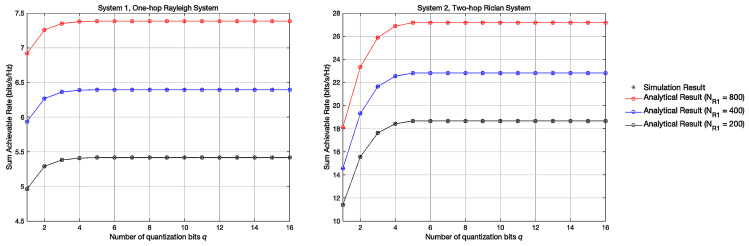
Achievable sum rate with different Quantization Bits *q*.

**Figure 5 entropy-23-01074-f005:**
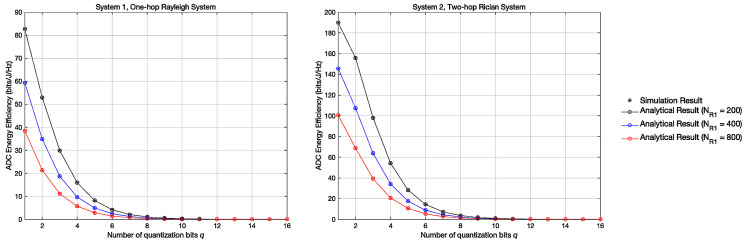
ADC Energy Efficiency with different Quantization Bits *q*.

**Figure 6 entropy-23-01074-f006:**
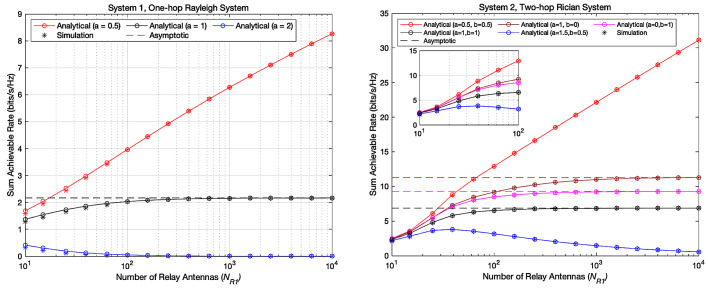
Asymptotic achievable sum rates with Different Scaling Indexes.

**Table 1 entropy-23-01074-t001:** Quantization distortion factor ρ under Different ADC quantization bits *q*. According to [10,16,17], the values of $\rho \;(\rho_{1},\rho_{2})$ when the number of quantization bits q≤5 is as follows.

*q*	1	2	3	4	5
$\rho ,\rho_{1},\rho_{2}$	0.3634	0.1175	0.03454	0.009497	0.002499

## Data Availability

1. The data in Table 1 is from paper [10,16,17]. 2. The data of energy loss formula is from [20,21]. 3. The data in simulation experiments is generated randomly by Matlab.

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
