# Peer review of "Low-Resolution ADCs for Two-Hop Massive MIMO Relay System under Rician Channels"

_entropy, 2021, doi:10.3390/e23081074_

Round 1
Reviewer 1 Report
The submitted paper presents a massive Multi-input Multi-output (MIMO) relay system capable of increasing the achievable sum-rate and energy efficiency. In particular, the analysis showed that the two-hop system has higher achievable sum-rate and energy efficiency than the one-hop system.
The paper is presents interesting results, but a clarification is needed.
During the paper it is not clear what is the considered antennas arrangement of the array. In fact, the antenna layout affects the massive MIMO performance [1]. For example, recently triangular lattice phased arrays showed improved massive MIMO performance with respect to square or rectangular lattice arrays [2]-[3] by enhancing spectral efficiency.
Figure 3 of the paper shows the achievable sum-rate in System 1 and System 2 as a function of the number of relay antennas (NR1). What is the array shape and the adopted lattice?
[1] X. Ge, R. Zi, H. Wang, J. Zhang, and M. Jo, ‘Multi-User Massive MIMO Communication Systems Based on Irregular Antenna Arrays’, IEEE Trans. Wireless Commun., vol. 15, no. 8, pp. 5287–5301, Aug. 2016, doi: 10.1109/TWC.2016.2555911.
[2] F. A. Dicandia and S. Genovesi, ‘Exploitation of Triangular Lattice Arrays for Improved Spectral Efficiency in Massive MIMO 5G Systems’, IEEE Access, vol. 9, pp. 17530–17543, 2021, doi: 10.1109/ACCESS.2021.3053091.
[3] F. A. Dicandia and S. Genovesi, ‘Spectral Efficiency Improvement of 5G Massive MIMO Systems for High-Altitude Platform Stations by Using Triangular Lattice Arrays’, Sensors, vol. 21, no. 9, p. 3202, May 2021, doi: 10.3390/s21093202.
Author Response
Dear reviewer,
Thank you very much for your question and suggestion for clarification. And also thank you for the recommended readings! It’s very meaningful to learn more about the influence of antennas arrangement on massive MIMO performance. In our paper, we discussed a more general situation. In the fast-fading matrix, we used d and θ to represent the distances and angles between each user, antennas and targets. In the final experiments (including Experiment 1, which produces Figure 3), we performed 1000 Monte Carlo simulations to eliminate the influence of random location distribution. We will set arranging the antennas as a future study topic to better our system. Thank you!
Bests,
Xinyi
Reviewer 2 Report
This paper examined building an effective massive Multi-input Multi-output (MIMO) relay system by increasing the achievable sum-rate and energy efficiency.
The work is well organized and appropriately carried out. I have looked at the mathematics and it looks sound. I would have liked to have seen a longer conclusion which discussed the effects of the various methods investigated were. At this p point, I recommend these parts to be revised again and be expressed more analytically in a revised version:
1)The study lacks a clear comparison between the submitted paper and the more relevant literature contributions, which should highlight the main advantages of the current submission.
2) please add a new section about the literature and highlight the contributions to your work.
3) How about latency. It will be good if you discuss the latency issue.
4) Figures resolution needs to improve.
5) What are the limitations of the proposed method?
6) How could/should futures studies improve the model?
7) English has to be improved to overcome some mistakes.
8) a minor comment, please check formatting according to the journal format.
Author Response
Dear reviewer,
Thank you very much for your review and suggestions! According to your suggestions, my clarification is as follows:
Reply to 1) and 2): We have refined and summarized the relevant literature contributions and organized them in 1.1 Related Works under 1. Introduction. Also, we’ve compared the contributions of our paper with the previous work and listed the main contributions of our paper in 1.2 Contributions.
Reply to 3): It’s a great suggestion and discussing latency is indeed a very important branch. We have also thought about this issue while we went over the setting of perfect Channel State Information (CSI), but in regard to the limitation of time and article length, we haven’t finished analyzing imperfect CSI. We have set it as our follow-up study direction and will carry on working on it.
Reply to 4): Thank you for your reminder, we have replaced our figures with the higher-resolution version.
Reply to 5) and 6): As is mentioned in your third suggestion, we haven’t analyzed the imperfect CSI setting. This results in certain limitations in the scalability of our model. In our future analysis, we will apply our model to the imperfect CSI and do the mathematical analysis there.
Reply to 7) and 8): Thank you for your reminder, we have checked out the grammar mistakes and refined our format.
Bests,
Xinyi
Round 2
Reviewer 2 Report
I am satisfied with the latest revision.